# The Emerging Role of Neutrophil Granulocytes in Multiple Sclerosis

**DOI:** 10.3390/jcm7120511

**Published:** 2018-12-03

**Authors:** Tonia Woodberry, Sophie E. Bouffler, Alicia S. Wilson, Rebecca L. Buckland, Anne Brüstle

**Affiliations:** The John Curtin School of Medical Research, The Australian National University, Canberra 2600, Australia; Tonia.Woodberry@anu.edu.au (T.W.); Sophie.Bouffler@anu.edu.au (S.E.B.); Alicia.Wilson@anu.edu.au (A.S.W.); Rebecca.Buckland@anu.edu.au (R.L.B.)

**Keywords:** multiple sclerosis, neutrophils, NETs, EAE, treatment

## Abstract

Multiple sclerosis (MS) is a demyelinating disease of the central nervous system with a strong autoimmune, neurodegenerative, and neuroinflammatory component. Most of the common disease modifying treatments (DMTs) for MS modulate the immune response targeting disease associated T and B cells and while none directly target neutrophils, several DMTs do impact their abundance or function. The role of neutrophils in MS remains unknown and research is ongoing to better understand the phenotype, function, and contribution of neutrophils to both disease onset and stage of disease. Here we summarize the current state of knowledge of neutrophils and their function in MS, including in the rodent based MS model, and we discuss the potential effects of current treatments on these functions. We propose that neutrophils are likely to participate in MS pathogenesis and their abundance and function warrant monitoring in MS.

## 1. Introduction

Neutrophils are a key population of granulocytes that are frequently involved in the initiation of an inflammatory response. They are produced in the bone marrow and have a half-life of hours to days [1,2]. Their characteristic multi-lobular nucleus, as well as the presence of granules arranged into organelles within their cytoplasm, explains their commonly used name polymorphonuclear leukocytes (PMN). Upon infection or injury, neutrophils begin a process of migration via chemotaxis, where they move along a chemical gradient in response to inflammatory signals [3]. Once neutrophils have entered the target site and been activated, they may carry out various effector functions to neutralize invading pathogens. The effector functions of neutrophils include phagocytosis, degranulation, and release of reactive oxygen species (ROS) [3,4].

The most recently described effector function of neutrophils is their ability to extrude DNA to produce neutrophil extracellular traps (NETs) in a process called NETosis [5,6]. The overall process of NETosis involves chromatin decondensation followed by the expulsion of DNA combined with antimicrobial factors from within the neutrophil [7]. The precise mechanism of NET release is still being investigated. Key mediators of DNA decondensation in this context include peptidylarginine deiminase 4 (PAD4) [7,8] and neutrophil elastase (NE) [9].

As the name suggests, the configuration of NETs facilitates their key function, which is to physically trap invading pathogens [5,10,11,12] while also increasing the dispersal of granular proteins—such as NE, myeloperoxidase (MPO), and cathepsin G—into the extracellular environment [5,10]. Neutrophils and NETs also assist the interaction between the innate and adaptive branches of the immune system, for example by activating antigen presenting cells [13,14] and therein promoting the differentiation of the inflammatory T helper 17 (Th17) cell population. Interestingly, Th17 cells are found in sites of acute inflammation and contribute to most autoimmune conditions [15]. Of note, their signature cytokine interleukin (IL)-17 is a key mediator of neutrophil recruitment [16].

While NETs have been most well-characterized during bacterial infections [17,18], including sepsis [19,20], where the web-like structures trap and neutralize invading bacteria, they have also been shown to be involved in the pathogenesis of atherosclerosis [14] and thrombosis [21]. Interestingly, they have been linked to autoimmune conditions including psoriasis [22], systemic lupus erythematosus (SLE) [23], and multiple sclerosis (MS) [24].

MS is a multi-faceted disease with a strong autoimmune, neurodegenerative, and neuroinflammatory component, affecting over 2 million people worldwide [25]. The disease is primarily diagnosed by the identification of plaques, characterized by demyelination and inflammation, throughout the central nervous system (CNS) [25]; however, further studies have been undertaken to identify additional hallmarks of early disease pathology, such as spinal cord atrophy or microstructural changes [26,27], retinal degeneration [28], and differences in thalamic volume [29]. Due to the variability of symptoms between MS patients, a standardized assessment tool was developed to quantify disability status associated with disease severity (expanded disability status scale—EDSS), with scores focusing on differences in movement [30].

Neutrophils have been extensively studied in the general context of neuroinflammation, for example in amyotrophic lateral sclerosis where it has been shown that neutrophil numbers are elevated and correlate with disease progression [31]; however, their specific role in MS is still not well defined. Much knowledge has been derived from animal models of experimental autoimmune encephalomyelitis (EAE). EAE mimics the autoimmune component of MS, and aspects of neuromyelitis optica spectrum disorders (NMOSD), and acute disseminated encephalomyelitis (ADEM) [32,33,34].

## 2. Involvement of Neutrophils in the EAE Model

In rodent EAE, neutrophil numbers are expanded in the periphery and the CNS before and during the onset of symptoms [35,36]. The pathogenic role of neutrophils in EAE has been confirmed through multiple studies. Depletion of neutrophils [13,35,37], inhibition of neutrophil migration by depletion of the neutrophil chemokine receptor CXCR2 [38,39], and depletion of neutrophil attracting cytokines—such as IL-17 [40] and granulocyte macrophage-colony stimulating factor (GM-CSF) [41]—significantly ameliorate the onset and severity of EAE. These studies have predominantly used the myelin oligodendrocyte glycoprotein 35–55 (MOG35–55) induced model of EAE [42]. This model is strongly dependent on Th17 cells [43,44] and both GM-CSF and IL-17 are derived from CNS infiltrating T cells [40,45], further underlining the strong interaction between this adaptive cell population and neutrophils in the MS model.

Neuroinflammation, as seen in MS and its corresponding animal model EAE, is based on a massive infiltration of immune cells into the CNS, which is normally a strongly controlled site of immune privilege. The blood brain barrier (BBB), which is one mechanism to prevent this infiltration, is impaired in EAE. One of the potential methods by which neutrophils contribute to EAE pathology is by facilitating this BBB breakdown, as depletion of neutrophils restores BBB integrity [35]. Recent studies in Alzheimer’s disease [46] and cerebral ischemia [47] discuss the possibility that the BBB breakdown is mediated through NETosis. Indeed, the cytotoxic effect of NETs has been clearly shown on human alveolar epithelial cells [48]. Studies depleting MPO, a protein abundant in NETs, attenuates EAE, and reducing reactive oxygen species (ROS), a product of neutrophils and the main intracellular driver of NETosis, re-established the BBB function [49,50,51]. Furthermore, inhibition of ROS production reduces the severity of symptoms and modulates the immune response in EAE [50,52]. Inhibition of NE, another neutrophil intrinsic enzyme involved in the processes leading to NETosis, reduces neutrophil infiltration into the optic nerves in EAE, also suggesting a role in MS and in particular NMOSD [53], where NE has already been implicated in disease progression [54].

The changes in EAE progression in response to altering neutrophils and their functions underline the importance of neutrophils in neuroinflammation.

## 3. Role of Neutrophils in MS

The role of neutrophils in people with MS remains unknown and research is ongoing to better understand the phenotype, function, and contribution of neutrophils in both disease onset and stage of disease. Past studies that have evaluated neutrophils in MS can be categorised as primary, secondary, or tertiary studies. Primary studies have directly counted neutrophils in blood, body tissues, or cerebrospinal fluid, and indirectly assessed neutrophils in plasma or serum. Secondary studies have characterized the phenotype and function of neutrophils, including the production of ROS and NETs, ex vivo. Tertiary studies have evaluated direct and indirect neutrophil modulation in vivo.

Only a few tertiary studies have been performed and these have evaluated the effects of blocking GM-CSF or the administration of granulocyte colony stimulating factor (G-CSF) in people with MS. GM-CSF blockade has been evaluated in a phase 1b study in a small group of MS patients, with intravenous infusion of the recombinant human monoclonal antibody (hMAb) MOR103, which binds GM-CSF [55]. MOR103 treatment-emergent adverse events were mild to moderate, occurring at comparable rates between the infused and placebo group. Study authors concluded that tolerability was acceptable and GM-CSF antibodies may provide a novel approach to MS therapies.

Exacerbation of MS was found upon G-CSF administration (high dose conditioning therapy) as part of immunoablation treatment, with neurologic worsening [56]. This first report that G-CSF was associated with increased inflammation was followed by a case report describing severe disease exacerbation with EDSS progression and several new brain lesions following G-CSF administration to a 31-year-old woman with previously stable RRMS, following treatment for breast cancer with chemotherapy and subcutaneous G-CSF [57]. Study authors postulated that G-CSF may contribute to a temporary flair of disease activity and challenged the use of G-CSF or GM-CSF in MS.

Secondary studies evaluating neutrophil phenotype and function in MS are prevalent and heterogeneous, producing no consensus as to the effects of disease on neutrophils or vice versa. Neutrophil oxidative burst studies either describe neutrophils as primed, exhibiting enhanced ROS production [24], or as normal, with similar ROS production, in healthy controls and people with MS, irrespective of treatment [58]. However, in this study, neutrophils from the treated group with normal oxidative burst demonstrated significantly reduced ability to kill bacteria (in vitro) [58]. Furthermore, in people with MS, with or without treatment, a slight reduction in neutrophil production of pro-inflammatory cytokines (IL-1β and IL-8) was found in response to bacterial stimulation (in vitro) [58].

Granulocytic myeloid derived suppressor cells (G-MDSCs), which are neutrophils with an immune suppressive function, have been detected in the blood of people with MS in higher abundance during active RRMS compared to the relapse recovery phase. Furthermore, G-MDSCs were elevated in people with MS compared to those without independent of disease status [59]. This first report of G-MDSCs in MS demonstrated suppression of the activation and proliferation of autologous T cells (in vitro), via mechanisms yet to be understood, and authors suggested G-MDSC may contribute to clinical recovery [59].

Blood phenotyping has described activated neutrophils in MS with: (1) increased surface expression of toll like receptor 2 and N-Formylmethionyl-leucyl-phenylalanine receptor [24,60]; (2) higher CD11b/CD18, CD10, and CD13 expression mainly in the course of disease exacerbation compared with remission [61]; (3) enhanced neutrophil protease activity [62]; and (4) neutrophil elastase (medullasin) activity [63]. Collectively, phenotypic studies suggest that activated neutrophils may participate in MS immunopathology. While priming prolongs the lifespan of PMN by activating anti-apoptotic signal transduction pathways and transcription factors to decrease transcription of pro-apoptotic factors, no change in apoptosis of neutrophils from people with MS was reported under stimulated or unstimulated conditions [58].

Molecular studies identified mRNA from the neutrophil-specific protein ASPRV1 in brain lesions, with higher amounts in severe MS compared to mild or moderate forms, and normal-appearing white matter [64]. While other studies report altered gene expression and provide indirect evidence of PMN in MS, they are not cited here due to the lack of neutrophil isolation or neutrophil specific gene expression.

Primary studies that directly or indirectly identify neutrophils support a role for neutrophils in MS. Neutrophils have been found in the cerebrospinal fluid in MS patients during relapse, at an early disease stage, with correlation between the cerebrospinal fluid neutrophils and IL-17A levels [65]. In particular, pediatric MS patients have neutrophils in the cerebrospinal fluid [66], and interestingly in adults the neutrophils in the cerebrospinal fluid tend to decrease with disease duration [65], suggesting activation of the innate immune system in early disease. Post-mortem CNS material obtained from an acutely ill MS patient revealed neutrophil infiltration associated with regions of BBB leakage [35]. In peripheral blood the neutrophil-to-lymphocyte ratio has been proposed as a marker of MS disease activity, with a higher ratio in MS, but with no association to progressive disease nor EDSS in a large cohort [67]. While other smaller cohort studies found an association between the neutrophil-to-lymphocyte ratio and EDSS score [68,69]. In plasma, measures of neutrophil associated factors NE, CXCL1, and CXCL5 correlate with MS lesion burden and clinical disability [36]. Another neutrophil product, MPO, is elevated in serum in MS [70]. Furthermore, MPO bound to DNA, which is considered a common marker of NETs, was found elevated in serum in MS, with no correlation to disease activity [24]. Despite mounting evidence for a role of neutrophils in MS we are yet to determine if they have a role in disease initiation, pathogenesis, and/or relapse.

## 4. Current MS Treatments and Their Effect on PMNs

There are a multitude of non-curative disease-modifying therapies (DMTs) licensed for MS (Table 1), and while none directly target neutrophils (or innate immunity), several DMTs do impact neutrophils. While prescription regimes differ between countries, in Australia all approved DMTs can be prescribed as a first line treatment for the more prevalent relapsing remitting form of MS (RRMS). In early 2018, Ocrelizumab became the first DMT showing some effectiveness in slowing the primary progressive form of MS (PPMS) [71]. Unfortunately, there are still no DMTs licensed for PPMS in Australia, underscoring the importance of continued research in this population.

In general, DMTs target aspects of a pathological process (directly or indirectly) to reduce neuroinflammation and attenuate disease severity. Therapies can be broadly categorized as immunomodulatory antibodies, immunomodulatory drugs, or cell cycle inhibitors.

## 5. Immunomodulatory Antibodies

The immunomodulatory antibodies Ocrelizumab [72], Ofatumumab [73], and Rituximab [74] target CD20 to deplete B cells, and all decrease neutrophil counts [75,76,77,78]. Alemtuzumab targets CD52 to deplete lymphocytes (B cells and T cells) [79], and impairs neutrophil function [80,81]. Natalizumab binds to the lymphocyte membrane integrin alpha-4 (α_4_) and competitively inhibits its interaction with vascular cell adhesion molecule (VCAM)-1, expressed on endothelial cells, thereby interfering with lymphocyte migration [82,83,84]. Natalizumab does not affect neutrophil numbers [85,86].

## 6. Immunomodulatory Drugs

The immunomodulatory drug Fingolimod is a sphingosine analogue that causes internalization of lymphocyte S1P receptors, leading to sequestration of lymphocytes in lymph nodes and other secondary lymphoid organs [87]. While initial use has no impact on neutrophil counts, chronic dosing leads to a decline in neutrophil count [88]. Glatiramer acetate, a synthetic peptide mimic of myelin basic protein that aims to inhibit T cell activation via competitive binding to MHC class II is not reported to impact neutrophils [89]. Interferon beta is a recombinant protein that has broad immune suppressive capability [90] and may reduce neutrophil counts [91]. Dimethylfumarate is also an immune suppressive [90,92,93] but may act to alter neutrophil function [94].

## 7. Cell Cycle Inhibitors

The cell cycle inhibitors Cladribine, Cyclophosphamide, and Mitoxantrone disrupt DNA synthesis of lymphocytes and other cells and each reduces neutrophil counts [98,99,100,105,106]. Teriflunomide is a pyrimidine synthesis inhibitor that is cytotoxic to rapidly dividing cells including lymphocytes and can cause neutropenia [108,109,110,111].

## 8. Discussion

While the view on the impact of neutrophils in neuroinflammation in animal models is sharpening, we remain mostly in the dark about their role in MS. The diversity of DMT formulations (comprising antibodies and chemicals) and varied mechanisms of action, highlight our broad and currently limited understanding of MS pathology. This lack of understanding extends to DMT prescription, as neurologists predominantly decide—case by case—which DMT is appropriate; considering safety, administration, patient compliance, efficacy, and cost. Despite this individually considered approach, DMTs can only be determined as efficacious, or not, following clinical monitoring for periods of months to years. Our current inability to predict or rapidly measure the effectiveness of any DMT is a limitation deserving of additional research, as this can impact patient wellbeing.

A comparison of the effects of all common treatments (Table 1) reveals the fact that most impact neutrophils in a negative way, by either reducing numbers or altering their function. The question remains whether this is just the fragile nature of neutrophils, or if effects on neutrophils might contribute to disease pathology. The proposed mechanism of action of these DMTs directly and indirectly on neutrophils is illustrated in Figure 1.

Interesting ideas about how neutrophils could contribute to disease pathology emerge from the above described rodent studies, one of these being the newly discovered NETs and their associated molecules. The cytotoxic role of NETs is well established and their potential role in the breakdown of the BBB has been postulated in other neuroinflammatory diseases [46,47]. While some human studies identified elevated NETs in people with MS [24], a strong functional link is still missing. One major problem in this regard is the technical challenge in detecting NETs [112]. The most established and commonly used methods are based on microscopy, which by itself provides some limitations when working with diseases of the CNS. Measurements in blood, plasma, or cerebrospinal fluid mainly use the detection of free DNA or DNA bound to molecules associated with NETs, such as MPO. While these could be considered direct measures of NETs, they also run the risk of providing false positives. For example, free DNA is released due to multiple processes such as tissue damage or necrosis and even dual detection of MPO and DNA cannot be directly linked to a neutrophil origin. Indeed, most molecules detected to determine neutrophil function are also produced by other cells. MPO, for example, is highly expressed by monocytes and macrophages, the chemokine CXCL2 was even first described as ‘macrophage inflammatory protein 2’ [113] and CXCL4 carries the name platelet factor 4 [114].

It is noteworthy, as mentioned above, that neutrophils are not a homogeneous population as they can differ in morphology and function [115]. Some studies even divide them into more inflammatory N1 and more suppressive N2 cells [116], while others attribute suppressive functions of neutrophils to G-MDSCs [117]. A recent multidimensional analysis based on proliferation capacity, transcription, and marker expression suggested functional differences are based on neutrophil maturation status rather than differentiation [118]. Whether different effects of neutrophils in MS can be attributed to different subpopulations or maturation states is yet to be clarified.

Furthermore, with increasing appreciation that neutrophils can influence adaptive immune cell activation [119] and potentially present antigens [120], comes the open question of whether neutrophils modulate T cells or B cells in MS. This along with the inverse question of whether neutrophils are modulated by other immune cells during disease and what the effects of current DMTs on neutrophils are (direct or indirect) remain to be determined.

## 9. Conclusions and Future Directions

Extensive studies using rodent models of MS and data obtained from other neuroinflammatory diseases strongly suggest a pathologic role for neutrophils in MS. The variety of human studies (primary and secondary) that have looked at neutrophils, and their function in MS, provide further evidence that these highly abundant cells are not bystanders to be overlooked in MS. However, the few tertiary studies focused on manipulation of neutrophils in vivo remind us to proceed with caution. We are just at the start of our exploration into neutrophil heterogeneity and are yet to understand how different subsets might bear different effector functions, including inflammatory and anti-inflammatory properties. We anticipate that one avenue of study likely to provide valuable insights is the manipulation of specific neutrophil effector functions. It will be imperative to single out pathologic effects and mitigate these rather than compromise the whole heterogeneous neutrophil population and in particular alter the many beneficial roles of neutrophils. Our current knowledge about neutrophil heterogeneity is restricted and even the detection of hallmark effects, like NETosis, is still in its infancy. Nevertheless, multidimensional and single cell analyses are developing fast and it is only a matter of time before these technical problems are solved.

Current research is actively investigating whether neutrophils, their subgroups, or specific effector functions can be used as biomarkers. Neutrophil manipulation may, in the future, aid MS treatments or even become the basis of new DMTs.

## Figures and Tables

**Figure 1 jcm-07-00511-f001:**
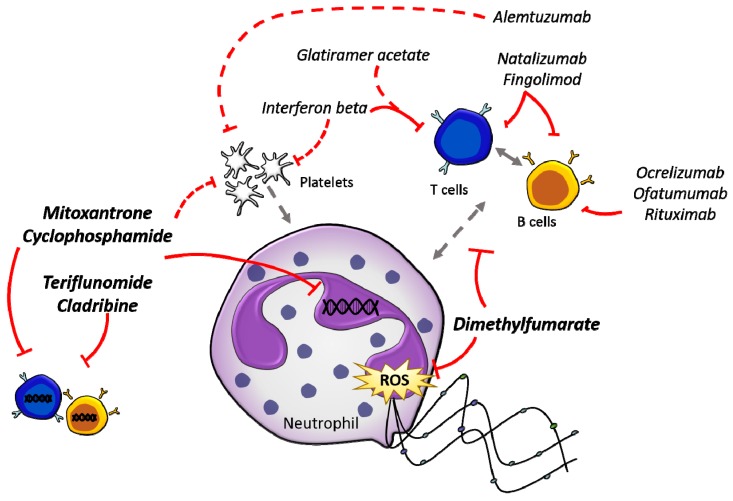
Direct and indirect effects of current DMTs on neutrophils. There are a multitude of DMTs (*italicised*) currently used in the treatment of MS, which target different components of the immune system. The direct interactions of DMTs with immune cells are indicated by solid red lines, while indirect interactions are indicated by dashed red lines. DMTs that have been shown to directly affect neutrophils are written in bold, while the other DMTs indirectly affect neutrophils through the modulation of other immune subsets. Grey arrows indicate direct (solid) and indirect (dashed) cell to cell interactions.

**Table 1 jcm-07-00511-t001:** Non-curative disease modifying therapies licensed for MS and their impact on neutrophils

Name	Short Description/Mechanism of Action	PMN Impact		Limitations/Risks and Non-Specific Affects
**Alemtuzumab** Lemtrada^®^/Campath^®^/Remniq	CD52 hMAb, depletes CD4/8 and B cells	**⬇**	[80,81]	Secondary B-Cell autoimmunities, infections, affects monocytes [79,95]
**Ocrelizumab** Ocrevus^®^	CD20 hMAb, depletes B cells	↓	[75]	IR, infections, increased risk of malignancies [71]
**Ofatumumab** Arzerra^®^/HuMax-CD20	CD20 hMAb, inhibits B cell activation	↓	[76]	IR, HBV reactivation, PML
**Rituximab** Rituxan/RTX	Chimeric CD20 MAb, depletes B cells (T cells)	**⬇**	[77,78]	IR (mild) [74]
**Natalizumab** Tysabri^®^	Alpha-4 integrin hMAb, interferes with lymphocyte migration	→	[85,86]	Increased risk of PML, fatigue and allergic reaction [82,96] affects monocytes, macrophages NK cells, DCs
**Fingolimod** Gilenya^®^	Sphingosine analogue, inhibits lymphocyte egress from secondary lymphoid organs	**↴**	[88]	Lymphopenia, effects on liver, heart, blood pressure, CNS (neural cells and glia), infections [97]
**Cladribine** LEUSTATIN^®^/MAVENCLAD^®^	Deoxyadenosine analogue which disrupts lymphocyte metabolism and DNA synthesis	**⬇**	[98]	Lymphopenia, infections, affects monocytes [98]
**Cyclophosphamide** Cytoxan^®^/Neosar^®^/Endoxan^®^	Alkylating agent (group of oxazaphosporines) targeting T cell interstrand DNA crosslinking	⤷⤴	[99,100]	Slight risk of blood cancer, cardio and bladder toxicity. Affects leukocytes, RBC and platelets. [99]
**Dimethyl fumarate (DMF, BG-12)** Tecfidera^®^	Methyl ester of fumaric acid, an immune suppressant acting on T and B cells	⇥	[94]	Flushing, abdominal pain, diarrhea, nausea, lymphopenia, increased risk of PML [93]
**Glatiramer acetate** Glatopa^®^/Copaxone^®^	Synthetic 4-mer peptide which acts as a mimic of myelin basic protein	?		Infusion reactions, depression, headache, nasopharyngitis, fatigue [89,101,102,103]
**Interferon beta** Avonex^®^/Betaferon^®^/Extavia^®^/Betaseron^®^/PLEGRIDY^®^/Rebif^®^	Interferon beta-1a/peginterferon beta-1a. Immune suppressant shifting from effector to suppressor phenotype	↓	[91]	IR, depression [104]
**Mitoxantrone** Novantrone^®^	Type- II topoisomerase inhibitor targeting B and T cell proliferation through inhibition of DNA synthesis and repair	**⬇**	[105,106]	Infections, heart damage, liver damage, birth defects, increased cancer risk, effects on platelets, complement mediated myelinolysis [105,107]
**Teriflunomide** Aubagio^®^	Pyrimidine synthesis inhibitor -Cytostatic or cytotoxic effect on rapidly dividing cells (T cells and B cells)	**⬇**	[108,109,110,111]	Diarrhea, nausea, hair thinning, higher aminotransferase levels (all dose dependent) [108]

hMAb, human monoclonal antibody; IFN, interferon; TNF, tumour necrosis factor; IL, interleukin; IR, infusion reaction; HBV, hepatitis B virus; PML, progressive multifocal leukoencephalopathy. ⬇ severe decrease in number (neutropenia); ↓ decrease in number; → unchanged number; ↴ decrease in neutrophil number following chronic dosing; ⤷⤴ initial reduction followed by recovery; ⇥ unchanged number but impaired function; ? unknown.

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
