# Peer review of "The Emerging Role of Neutrophil Granulocytes in Multiple Sclerosis"

_jcm, 2018, doi:10.3390/jcm7120511_

Round 1
Reviewer 1 Report
This is an interesting review that addresses a timely issue. Our knowledge of the role of neutrophils in demyelinating diseases is limited. While the review is quite comprehensive one is left with an unsettled feeling that authors do not want to any conclusions regarding the role of neutrophils in pathogenesis of disease. The review would be more interesting with some speculation.
The section of DMT seems in the wrong place. I understand the authors want to discuss possible effects of DMT on neutrophils and this is a good idea - but some lead into the subject would help.
Author Response
We would like to thank the reviewer for their insightful comments.
We agree that our position is not very speculative but we ourselves strongly believe that we need more investigations to make a call. We made some minor changes in the abstract to accommodate for the lack of this.
To fit the section about DMT's better into the flow of the text we added a few words at the end of the previous and the beginning of the section.
All changes are marked by "track changes" in the document attached.
Reviewer 2 Report
This is a comprehensive summary of the sparse knowledge we have on the role of neutrophils in multiple sclerosis. The paper is well written, I have only a few suggestions:
- abstract: MS is not only a neuroinflammatory but also a neurodegenerative disease, please rephrase accordingly
- p. 2 line 49 on various diseases, here or elsewhere NMO should be mentioned including the applicable publication by Saadoun et al.,PMID:22374891
- p2 line 53: this is not the full story, neurodegeneration occurs in MS from earliest disease stages, see and add refs PMID:29359174,PMID:29435472,PMID:28761906,PMID:23702433
- p. 2 line 58: this needs to be explained in more clarity, EAE is not the animal model for NMO and ADEM
- p. 3 line 103: are there subsequent studies planned, underway, published?
- please correct typo in figure 1: it should read dimethyl fumArate
- see and add this paper on NAT: PMID:26140281
- also ALS and neutrophils could be briefly mentioned:PMID:27308304
Author Response
We would to thank this reviewer for the helpful and appropriate comments.
We added the aspect of neurodegeneration through out the manuscript and included all the citations suggested.
The changes are included in the attached document indicated by "track changes".
About subsequent studies on GM-CSF, we contacted the authors but until now did not get a response. However there is no additional trail in MS on the clinical trial registry.
We hope these changes are satisfactory.